# Characterization of a Human Platelet Lysate-Loaded Keratin Hydrogel for Wound Healing Applications In Vitro

**DOI:** 10.3390/ijms23084100

**Published:** 2022-04-07

**Authors:** Kameel Zuniga, Alisa Isaac, Sean Christy, Nicole Wrice, Lauren Mangum, Shanmugasundaram Natesan, Luke Burnett, Robert Christy, Christine Kowalczewski

**Affiliations:** 1Combat Wound Care, US Army Institute of Surgical Research, JBSA Fort Sam Houston, San Antonio, TX 78234, USA; kameel.m.zuniga.ctr@mail.mil (K.Z.); alisaisaac@tamu.edu (A.I.); seanec3@gmail.com (S.C.); nicole.l.wrice.civ@mail.mil (N.W.); soupamhere@gmail.com (L.M.); shanmugasundaram.natesan.ctr@mail.mil (S.N.); robert.j.christy12.civ@mail.mil (R.C.); 2KeraNetics, Inc., Winston-Salem, NC 27101, USA; lburnett@keranetics.com

**Keywords:** human-platelet lysate, keratin, hydrogel, injury, drug delivery

## Abstract

One of the promising approaches to facilitate healing and regenerative capacity includes the application of growth-factor-loaded biomaterials. Human platelet lysate (hPL) derived from platelet-rich plasma through a freeze-thaw process has been used as a growth factor rich therapeutic in many regenerative applications. To provide sustained local delivery of the hPL-derived growth factors such as epidermal growth factor (EGF), the hPL can be loaded into biomaterials that do not degrade rapidly in vivo. Keratin (KSO), a strong filamentous protein found in human hair, when formulated as a hydrogel, is shown to sustain the release of drugs and promote wound healing. In the current study, we created a KSO biomaterial that spontaneously forms a hydrogel when rehydrated with hPL that is capable of controlled and sustained release of pro-regenerative molecules. Our study demonstrates that the release of hPL is controlled by changing the KSO hydrogel and hPL-loading concentrations, with hPL loading concentrations having a greater effect in changing release profiles. In addition, the 15% KSO concentration proved to form a stable hydrogel, and supported cell proliferation over 3 days without cytotoxic effects in vitro. The hPL-loaded keratin hydrogels show promise in potential applications for wound healing with the sustained release of pro-regenerative growth factors with easy tailoring of hydrogel properties.

## 1. Introduction

Civilian and military open traumatic wounds and burns are often complex and non-uniform injuries leading to complications in wound management. In order to optimize the healing process, point of injury treatments are currently being developed which are easily applied, configure to wound dimensions, and stimulate tissue repair. Human blood plasma and derivatives are used as therapeutics for tissue repair from single or pooled donors [1]. The collected blood can be fractionated into different components, including platelet rich plasma (PRP), that can be used to treat numerous subacute and chronic tissue injuries [2,3,4,5,6,7,8]. When tissue is wounded, platelets secrete proteins that form a hemostatic plug which then releases a multitude of growth factors attracting surrounding cells and aids in the repair of the wounded tissue [9,10,11,12]. Although PRP gels are shown actively to promote angiogenesis [7,12,13], fibroblast proliferation, and overall re-epithelialization [12,14,15], clinical outcomes are inconsistent because the PRP preparation procedures differ and are patient-dependent due to varying platelet and chemokine concentrations [8,16]. To overcome this issue, human platelets are lysed to extract the components within the platelets (hPL), including pro-regenerative growth factors, through a freeze-thaw process [17]. In addition, hPL can be stored at −80 °C, unlike plasma, which has a shelf life of only 5 days. Studies show that hPL increases angiogenesis, cell growth, and adhesion involved in the wound healing process by stimulating cells associated with skin (fibroblasts and keratinocytes) through growth factors’ release [18,19,20,21].

Hydrogels loaded with growth factors are used as therapies to achieve regeneration of the functional tissue and improve the overall wound healing outcomes. Hydrogel-based therapies are popular due to their ease of use, ability to contour complex wound geometries, and attain desired physical and biological properties for cell growth and tissue regeneration [22,23]. Synthetic hydrogels are an attractive option, since their chemical properties can be tailored to achieve a desired functional outcome [24,25]. However, synthetic hydrogels lack the cell binding motifs to promote cell attachment and proliferation and may elicit a foreign body response, usually requiring further modification by crosslinking cell-binding peptides [26,27,28]. Naturally based hydrogels are popular due to their biocompatibility and inclusion of binding motifs that increase attachment and growth of cells without eliciting a foreign body response [29,30,31,32,33].

Keratin-based biomaterials are used in tissue engineering and regenerative medicine (TERM) applications to improve overall wound healing outcomes [33,34,35,36,37,38]. Keratin is an intermediate filament protein naturally found in epithelial tissue as well as feathers, hooves, wool, and human hair. Keratin protein contains cell-binding motifs, such as leucine-aspartic acid-valine (LDV), which is shown to promote cell attachment and proliferation [33,38,39,40]. Keratin can be extracted by oxidation (keratose), and its properties can be tuned to meet the specific needs of the biomaterial scaffold. Keratose is non-disulfide crosslinked, leading to their higher water solubility and hydrolytic degradation rate of keratin [34,37,38]. Keratose hydrogels were previously shown to demonstrate sustained release of drugs with controlled degradation [35,37,38,41,42,43]. This sustained release of drugs and growth factors are important in TERM applications so that the therapeutic in particular meets the time of action required for full regeneration of the damaged site [25]. If the loaded biomaterial/scaffold degrades too rapidly in vivo, which is often associated with collagen hydrogels, sustained release of the drug or growth factor is not met for enough time, and the therapeutic will not reach full therapeutic potential [35,44,45,46].

In this initial in vitro study, keratose derived from human hair, referred as keratin (KSO), was rehydrated with varying concentrations of hPL to form 15, 22.5, and 30% weight/volume (*w*/*v*) hydrogels. The capability of a hPL-loaded hydrogel to sustain the release of hPL pro-regenerative molecules, specifically epidermal growth factor (EGF), was investigated. Furthermore, the storage modulus was determined through rheological oscillation, and hydrogel structure and porosity were characterized through scanning electron microscopy. Total protein release and EGF were measured to quantify hydrogel degradation and growth factor release, respectively. Cell proliferation of human dermal fibroblasts (hDFs) over 3 days was measured in vitro to determine whether the hydrogel promoted cell growth. We were able to demonstrate that hPL-loaded KSO hydrogels were able to support sustained release of hPL without any cytotoxic effects in vitro. This initial in vitro study with cells grown in 2D demonstrated the capability of hPL-loaded KSO hydrogels to support proliferation of skin-specific cells.

## 2. Results

### 2.1. Mechanical Properties of Hydrogels

Rheological testing of hydrogels was conducted to determine the storage modulus and stability of hydrogels with varying keratin (KSO) and human platelet lysate (hPL) concentrations. As shown in Table 1, an increase in KSO concentration increased the storage modulus 3.9 to 4.5-fold from 15 to 22.5% *w*/*v* and 3.3 to 5.5-fold from 22.5 to 30% *w*/*v* KSO. On the other hand, increasing the hPL-loading concentration from 0 to 100% decreased the storage modulus by 2.2, 1.9, and 1.2-fold for 15, 22.5, and 30% *w*/*v* KSO, respectively. The tan δ of each hydrogel was less than <1, which indicates that the hydrogel exhibited more elastic than plastic behavior. However, increasing the KSO concentration slightly increased the plasticity of the hydrogel as well. Increasing the hPL concentration did not increase the tan δ. Further, the complex modulus was linear for all hydrogel formulations (data not shown), indicating that the hydrogels were stable.

### 2.2. Hydrogel Structure and Porosity

To observe the KSO and hPL concentration effects on hydrogel structure and porosity, SEM images were compared between varying KSO concentrations loaded with 0% hPL and varying hPL concentrations with 22.5% KSO hydrogels (Figure 1). SEM images show a porous architecture for 15, 22.5 and 30% KSO hydrogels with an increase in KSO concentration leading to an observable decrease in pore size with increasing *w*/*v* (Figure 1A). Images also confirmed the formation of porous hydrogels with an increase in hPL as 22.5% KSO hydrogels were reconstituted with an increasing hPL concentration (Figure 1B). The increasing hPL concentration showed no observable changes in pore size and porosity. Although an increase in hPL concentration led to significant mechanical changes, differences at the macroporous level were not observed.

### 2.3. Cumulative Total Protein Release from Hydrogels

We sought to determine whether total protein release could be controlled by varying the hydrogel formulation (both KSO and hPL-loading concentration). The protein release from each hydrogel was measured and plotted cumulatively over the course of 7 days. This cumulative protein release includes KSO and hPL that was released from the hydrogel. As shown in Figure 2, hydrogels loaded with 100% hPL had the highest cumulative protein release, with the lowest protein release observed with 0% hPL hydrogels. Therefore, the higher the hPL concentration, the higher the protein release was observed. This relationship was true for all KSO hydrogels. Similarly, cumulative protein release also increased with an increase in KSO concentration, with the highest release observed with 30% KSO hydrogels (Figure 3). From Figure 2 and Figure 3, the addition of hPL to the KSO hydrogel was shown to be more significant in changing the protein release profile of the hydrogel rather than increasing the KSO concentration. As shown by Figure 3A, when loaded with no hPL, KSO concentration has a greater effect on the protein release profile. However, with an increasing hPL concentration (Figure 3B–D), the change in KSO concentration does not change the cumulative release profile significantly.

### 2.4. EGF Release as a Measurement of hPL Release

Control of the drug and growth factor release is important in designing hydrogel biomaterials for regenerative medicine applications. Thus, determining whether the sustained release of hPL could be controlled by varying the KSO and hPL-loading concentration was important in designing this biomaterial. Human platelet lysate contains a number of pro-regenerative growth factors, amongst which epidermal growth factor (EGF) is a key effector protein that promotes stem cell proliferation to be present at signification levels in hPL. We chose EGF as a representative analyte to determine the release of pro-regenerative factors from hPL loaded KSO hydrogels [47]. The sustained release of EGF from hPL was measured over the course of 7 days (Figure 4). As shown by Figure 4A, 15% KSO hydrogels loaded with 50, 75, and 100% hPL released 350, 450, and 620 pg/mL, respectively (1.77-fold increase from 50 to 100% hPL). A similar pattern was observed with 22.5% KSO (260, 400, and 520 pg/mL for 50, 75, and 100 hPL, respectively) and 30% KSO (250, 300, and 400 pg/mL for 50, 75, and 100 hPL, respectively), in which increasing hPL concentration from 50 to 100% hPL increased the release of hPL by 2.00 (22.5% KSO) and 1.60-fold (30% KSO). By day 7, 15% KSO 100 hPL-loaded hydrogels had the highest EGF release (900 pg/mL) and 30% KSO 50 hPL-loaded hydrogels had the lowest EGF release (320 pg/mL). Significant differences were observed between all formulations at each time point.

Comparisons were also made between different KSO concentrations at each hPL loading concentration (Figure 5). Although the change in KSO concentration also affected the release, increasing the KSO concentration was not as significant as the increase in hPL concentration. For 50% hPL-loaded hydrogels at day 3, there was only a 1.56-fold increase from 15% to 30% KSO hydrogels. For 75 and 100% hPL-loaded hydrogels, there was only a 1.71 and 1.54-fold increase, respectively, from 15% to 30% KSO hydrogels. Overall, increasing the hPL-loading concentration and decreasing the KSO hydrogel concentration increased the EGF release, and therefore the hPL release.

### 2.5. Cell Proliferation of hDFs

To determine the effects of hPL-loaded KSO hydrogels on cellular proliferation, hDFs were treated with 15% KSO hydrogels loaded with varying concentrations of hPL as a proof-of-concept. An untreated group served as a control. The 15% KSO hydrogels were used because they were stable enough to form hydrogels and demonstrated the most varying protein and hPL release profile with sustained release (Figure 2, Figure 3, Figure 4 and Figure 5). As shown in Figure 6, an increase in cell proliferation was observed in all groups over the course of 3 days. No significant differences were observed at day 1 between all groups, which had similar normalized cell proliferation as the negative control (No KSO 0 hPL) and positive control (No KSO 100 hPL). By day 2, significant decreases in cell proliferation were observed for 15% KSO 0 hPL, 15% KSO 50 hPL, and 15% KSO 100 hPL when compared to 100 hPL only (*p* < 0.05). hDFs had the highest normalized cell proliferation at day 3, about 2-fold higher than day 1 for all samples. On day 3, there was no significant difference between hDF proliferation of 15% KSO hydrogels when compared to the positive control. Significantly lower cell proliferation was only observed with 15% KSO, 0 hPL hydrogels when compared to no treatment on day 3. Although no significant growth was observed between the 15% KSO hydrogels at different hPL loading concentrations, there was an increasing trend in cell proliferation as the hPL loading concentration increased on each day, indicating that the cell growth can be controlled by the release of hPL by changing the loading concentration.

## 3. Discussion

Many drug delivery biomaterial therapies are studied and developed to aid in healing and tissue regeneration of injuries. Notably, it is critical to develop biomaterials that are able to promote the sustained release of growth factors and drugs, due to growth factors having a short life-span and being unstable in vivo [48], and the long healing time of injuries. Therefore, action time of the growth factor needs to be extended to maintain proper growth factor concentrations at the wound site. Keratin (KSO) has been shown to maintain sustained release of drugs including ciprofloxacin and key growth factors such as VEGF and FGF [35,37,38,41,42,43]. In this study, KSO was used to create mechanically stable hydrogels that promoted the sustained release of active therapeutic molecules present in hPL and supported cell proliferation as a potential biomaterial that can be translated clinically for injury treatment.

Human platelet lysate-loaded KSO hydrogels were prepared and characterized for mechanical stability. Results from rheological studies confirmed stable viscoelastic hydrogels that were fabricated for all formulations. This was confirmed by the linear plot for the complex modulus and G’ being greater than G” for all hydrogel formulations. In particular, increasing the KSO concentration increased the G’ (storage modulus), whereas increasing the hPL-loading concentration decreased the G’. The increased strength observed with higher KSO concentration hydrogels is likely due to polymer chain entanglement, due to the elimination of free thiols of the oxidized keratin (keratose) used in this study [49]. Previous studies observed similar results in which keratin hydrogels with half the concentration having two to three orders of magnitude decrease in G’ [35]. In addition, the electrostatic and hydrophobic interaction may also have contributed to the formation of larger networks through the positive head and the negative domains of dimers possibly associating to form tetramers [50,51] and hydrophobic regions of the coil regions aggregating to increase the polymer molecular weight and hydrogel formation [51,52]. The decrease in G’ with an increase in the hPL-loading concentration may be a consequence of the variable composition of hPL, leading to nonuniformity and lowered strength. Similar results were observed in previous studies, in which an increasing hPL concentration leads to a decrease in mechanical properties [21,53,54]. Although the formulations were able to produce hydrogels spontaneously with signs of a stable network, they were still flowable and easy to work with, which is important for injectable biomaterials in vivo to fit the shape of the defect or injury.

Differences in hydrogel pore size were observed when the KSO concentration was varied. With an increasing KSO concentration, porosity of the hydrogels decreased due to higher protein density and chain entanglements. Electron microscope images corroborated to the data obtained by rheology, in which increases in the KSO concentration led to an increase in G’. Porosity and pore architecture of hydrogels play a significant role in the mechanical stiffness of hydrogels, in which increasing porosity causes the decrease in stiffness of the hydrogel [55]. However, an increase in hPL led to no observable differences in pore size, as observed in previous studies involving hydrogels loaded with hPL [54]. No precipitates were observed after processing conditions for SEM preparation, indicating that hPL was effectively loaded in the KSO hydrogels.

The protein release profiles from hPL-loaded KSO hydrogels include total protein content contributed by both KSO degradation and hPL that was released. The protein content released was highest within the first 24 h for all hydrogel formulations. However, rapid burst and degradation were not observed up to 7 days. After the first 24 h, the release profile becomes linear for all hydrogels up to 7 days, indicating sustained protein release and KSO hydrogel degradation. One of the limitations of this study is we were unable to distinguish between hPL release and total KSO hydrogel degradation; KSO hydrogels without hPL showed increasing protein release (KSO only) with a decreasing KSO concentration by almost 1.5-fold from 15% to 30% KSO. Previous studies on keratin hydrogels demonstrated that increasing the weight percentage decreases degradation [56]. This is most likely due to the higher degree of KSO chain entanglements and hydrophobic and electrostatic interactions with a higher KSO concentration [35,38]. These results correspond to the mechanical stability of the hydrogels, in which an increase in KSO concentration led to an increase in G’. An increase in hPL payload in the hydrogel resulted in increased total protein release. Santo et al. previously reported higher protein release from hPL-loaded methacrylated gellan gum hydrogels when loaded at higher concentrations [21]. While it is difficult to determine whether an increase in protein release is due to KSO degradation or higher hPL protein concentration and release, it is important to note that the mechanical stability of hydrogels decreased due to an increase in the hPL-loading concentration. Therefore, it is possible that an increase in hPL concentration led to lower mechanical stability, leading to higher KSO degradation.

In this study, we chose EGF as a representative growth factor to determine the hPL release from KSO hydrogels. Similar to the protein release results, an increase in KSO and hPL-loading concentration led to a decrease and increase in EGF release, respectively. A sustained release of EGF was observed until 7 days. The EGF release rate increased release with an increased hPL-payload. Still, no rapid burst of release was observed within the initial 24 h. The total EGF release increased after each time point and did not plateau after 7 days. Therefore, the release of hPL could possibly extend for even longer time points. By controlling the KSO hydrogel concentration, the release rate and concentration of essential growth factors can be easily tuned, unlike the simple diffusion delivery process from a vehicle. Further, the KSO hydrogels are not affected by enzymatic degradation due to the lack of any inherent enzymes that break down keratin, unlike other major proteins, such as collagen and associated native ECM proteins that degrade rapidly in vivo by collagenase [35]. Therefore, we were able to demonstrate in this study that the sustained release of growth factors from KSO hydrogels was dependent on the rate of hydrogel hydrolytic degradation over time.

The role of tissue engineering is to support the growth and recruitment of cells to the injured area and regenerate new, healthy tissue. To determine cytotoxicity and whether hPL-loaded KSO hydrogels were able to promote cell growth, we treated hDFs with 15% KSO hydrogels over the course of 3 days in a two-dimensional in vitro culture. We decided to use 15% KSO hydrogels since this is the lowest concentration of KSO that has been shown to form a hydrogel spontaneously [43] and had the most desirable sustained release profiles based on the hPL-loading concentration. All groups, including no treatment and free hPL, supported the proliferation of the hDFs over the course of 3 days. Although there was a significant decrease in cell proliferation of hPL-loaded KSO hydrogels when compared to free-hPL, cell proliferation was not significantly different from no treatment, suggesting no cytotoxic effects from the KSO hydrogel. The lower cell proliferation observed with hPL-loaded KSO hydrogels when compared to free-hPL may be a consequence of sustained mitogen released from hydrogels, wherein wells supplemented with hPL were exposed to a bolus of growth promoting factors. This is important to note because, in vivo, scaffolds and other biomaterials cannot be replaced while conducting long-term studies. In addition, the free delivery of hPL may expose key growth factors susceptible to rapid enzymatic degradation.

Keratin is an attractive biomaterial for tissue engineering and regeneration purposes due to its unique properties of self-assembly into fibrous and porous scaffolds [57,58] and presence of cell binding motifs including leucine-aspartic acid-valine (LDV) and glutamic acid-aspartic acid-serine (EDS), which is capable of promoting cell attachment and proliferation [59,60]. For example, KSO hydrogels are shown successfully to regenerate nerve damage by inducing Schwann cell proliferation and migration in a mouse model [42]. In a more relevant model to our study, our group studied KSO hydrogels loaded with ciprofloxacin for tissue regeneration and prevention of methicillin-resistant *Staphylococcus aureus* (MRSA) and *Pseudomonas aeruginosa* (*P. aureus*) infection after an excision or burn wound in a porcine model [37,61]. These studies showed that the KSO hydrogels also supported sustained release of ciprofloxacin with infection prevention and supported wound healing. Keratin-based dressings are also used to facilitate wound healing and re-epithelialization while preventing a severe immune response in vivo [62,63,64]. We demonstrated in the present study similar results, with sustained release of hPL which supported the proliferation of hDFs without any cytotoxic effects.

There are a few limitations to this study that need to be noted in relation to future potential in vivo studies and clinical translation. Cells grown in a 2D environment behave differently than in vivo, a 3-dimensional (3D) environment. Although the primary purpose of this study was to determine whether hPL-loaded KSO hydrogels supported sustained release of growth promoting factors/mitogens present in hPL, we note that growing the cells on top or within the hydrogels may be more representative of 3D architecture in vivo. We also note that growth factor and total protein release (degradation) and cell proliferation studies need to be extended longer. The cumulative EGF and total protein release warrant longer time determination to understand the extended-time release beyond the plateau levels. In addition, tracking cell proliferation studies for extended time may help to determine the difference in cell-growth kinetic under the influence of hPL delivered through a KSO hydrogel. Despite these limitations, the results of this study show that KSO hydrogel was able to sustain the release of growth promoting factors from hPL hydrogel and that formulation parameters could easily be tuned by controlling hPL-loading and KSO hydrogel concentration. Further, KSO hydrogels were able to support increased cell growth over time. In summary, hPL-loaded KSO hydrogels show great promise to future translation in clinical applications involving injury treatment.

## 4. Materials and Methods

### 4.1. hPL Solution Preparation

Good manufacturing practice (GMP) grade human platelet lysate (hPL) was purchased from Cook Medical (Stemulate; Cook Regentec, Indianapolis, IN, USA). Frozen hPL was first thawed for 10 min in a 37 °C water bath. Solutions of different concentrations (0, 50, 75, and 100% of total volume) were prepared by diluting with sterile PBS. For the remainder of this study, 0, 50, 75, and 100% hPL will be referred to as 0 hPL, 50 hPL, 75 hPL, and 100 hPL, respectively.

### 4.2. hPL-Loaded Keratin Hydrogel Fabrication

Keratin (KSO) was obtained by oxidative extraction of human hair and purified using a patented and controlled manufacturing process by KeraNetics, LLC (Winston-Salem, NC, USA) [42,43]. Frozen KSO powder was first equilibrated at room temperature for 30 min. To fabricate 15, 22.5, and 30% *w*/*v* keratin hydrogels, 150, 225, or 300 mg of keratin were added to 1 mL of the respective solution (0 hPL, 50 hPL, 75 hPL, 100 hPL). The mixture was vortexed immediately for 30 s or until fully homogenous. Once mixed, the mixtures were centrifuged for 2 min at 2000 rpm at room temperature to remove any air bubbles. Each sample of hPL-loaded keratin hydrogel was transferred to a 1 mL syringe and was then capped and incubated at 37 °C overnight to polymerize.

### 4.3. Rheology

Hydrogels were prepared into silicone molds (25 mm × 1.5 mm) and incubated overnight at 37 °C as described previously in Section 2.2 in order to confirm stable hydrogel formation. To test for stability and material properties, samples were loaded onto a rotational rheometer (TA Instruments, DHR-3, Newcastle, DE, USA) with parallel plate geometry (1 mm gap width). The linear viscoelastic region (LVER) was determined through a dynamic oscillation at 1 Hz, and a frequency sweep from 0.01 to 10 Hz was performed at a constant stress within the LVER region. Each sample was analyzed in triplicate.

### 4.4. Scanning Electron Microscopy

Hydrogels were formed as previously described in Section 2.2. After incubation at 37 °C, hydrogels were frozen overnight at −80 °C in individual wells in a 12-well plate. Frozen hydrogels were lyophilized overnight (Labcon, Petaluma, CA, USA), mounted onto specimen stubs with carbon tape and paint, and gold sputtercoated. Samples were imaged using a Carl Zeiss SigmaVP (Oberkochen, Germany) scanning electron microscope at 300× magnification to visualize the hydrogel structure.

### 4.5. Protein Release

Protein release was measured by quantifying the total protein released at preassigned time points from 1.5 h to 7 days. A total of 250 µL of each hydrogel group were injected to the bottom of a 1.5 ml microcentrifuge tube and centrifuged at 2000 rpm for 2 min and incubated overnight at 37 °C. After incubation, 250 µL of sterile PBS were added on top of the hydrogel and incubated at 37 °C for the various time points up to 7 days. At each time point, 250 µL of PBS were collected and frozen at −80 °C, and subsequently the tube was replaced with 250 µL of fresh PBS and incubated until the following time point. Each condition was performed in triplicate, and the experiment was repeated three times. Cumulative protein was quantified by the Pierce BCA Protein Assay kit (ThermoFisher Scientific, Waltman, MA, USA) by diluting the collected samples with PBS by 1:10 or 1:12 dilution. Samples were prepared according to the BCA assay protocol, plated onto a 96 well plate, and absorbance was measured at 562 nm using a Synergy Mx plate reader (Biotek, Winooski, VT, USA). The total protein of each sample was approximated using the albumin standard curve.

### 4.6. EGF Quantification

Aliquots of samples collected at various time points (Section 2.5) were utilized to quantify hPL growth factor release. The Bio-Plex Pro Human Cancer Biomarker Panel 2, 18-plex standard (BioRad, Hercules, CA, USA) was used to quantify EGF release following manufacturer’s instructions. Samples were analyzed using a Bio-Plex 200 System (BioRad, Hercules, CA, USA). Cumulative EGF release was calculated over the one-week experiment.

### 4.7. Cell Culture

Primary hDFs were isolated (N = 3 subjects) and cryopreserved from dermal tissue obtained from de-identified human abdominal skin and underlying adipose tissue obtained from elective abdominoplasty surgeries with appropriate written consent. This study was conducted under a protocol reviewed and approved by the U.S. Army Medical Research and Materiel Command Institutional Review Board and in accordance with the approved protocol (H-11-020/M-10128; initial approval 27 June 2011). Briefly, the dermal tissue was separated from the skin samples from which the hypodermis was removed using sterile scissors, and the epidermis was removed by incubating the tissue in dispase at 37 °C for one h. The isolated dermal tissue was then finely minced with sterile scissors in Hanks’ balanced salt solution (HBSS; Invitrogen, Carlsbad, CA, USA). Samples were digested on an orbital shaker in HBSS with collagenase II (Life Technologies, Carlsbad, CA, USA) to a final concentration of 1 mg/mL at 37 °C. The digested cell suspension was filtered using 100 and 70 µm mesh filters (Corning, Corning, NY, USA). The resulting hDFs were plated, expanded in alpha minimal essential media (A-MEM) (Gibco, Waltham, MA, USA), at 37 °C in a in 5% CO_2_ environment until 70% confluent. The primary hDFs were cryopreserved and transferred to liquid nitrogen for long-term storage. For experiments, hDFs were thawed and cultured in T-150 flasks in A-MEM (phenol-free) media supplemented with 10% fetal bovine serum (FBS) and 1% antibiotic antimycotic (ABAM) solution and expanded to P2 or P3. Three technical triplicates per isolates (three separate donors were used) were used per experiment. hDFs were maintained in 5% CO_2_ atmosphere at 37 °C in a sterile cell-culture incubator. When hDFs reached 80% confluence, cells were detached by rinsing with PBS and subsequently treated with 0.25% trypsin for 3–5 min at 37 °C. The cell solution was then neutralized with media, collected, and centrifuged at 1900 rpm for 5 min in a 15 mL conical tube. The cell pellet was isolated from the supernatant and resuspended in fresh media. A cell solution of 0.8 × 10^6^ cells/mL was prepared, and 500 µL were seeded in 6 well plates to reach a seeding density of 0.4 × 10^6^ cells/well. A total of 2.5 mL of fresh media were subsequently added to each well and placed in a cell incubator at 37 °C overnight.

### 4.8. hPL Solution and hPL-Loaded Keratin Hydrogel Treatment

Following overnight incubation of hDFs (less than 17 h), cells were treated with either hPL solutions or hPL-loaded keratin hydrogels with pre-assigned conditions as stated in Section 2.2. The media was replaced with 3 mL of fresh media, and a 3 µm-pore size cell culture transwell insert was placed in each well. For non-hydrogel treatment groups, 25 uL of either 100, 75, 50, or 0% hPL was added to each well insert. For keratin hydrogel treatment groups, 25 µL of each respective hPL concentration loaded into a 15% keratin hydrogel were added into each individual insert. Subsequently, 1.475 mL of fresh media were added to each insert to bring the total volume to 1.5 mL. All sample plates were incubated at 37 °C and maintained at a 5% CO_2_ atmosphere in a sterile cell-culture incubator for 24, 48, or 72 h, at which time samples were collected. Triplicates were collected at each time point.

### 4.9. Cell Proliferation Assay

At each preassigned time point, the 6 well plate was inverted and gently flicked to remove as much media as possible without dislodging the cells and then gently blotted with a kim wipe. The plate was then covered, frozen, and stored at −80 °C until the assay was performed. To determine the rate of proliferation, the CyQuant Cell Proliferation Assay (ThermoFisher, Waltham, MA, USA) kit was utilized according to the manufacturer’s protocol. All plates for all time points as well as CyQuant kit reagents were removed and equilibrated at room temperature. First, a cell lysate solution was prepared by diluting the cell lysate stock buffer in deionized (DI) water at a 1:20 dilution. The CyQuant working solution was prepared by diluting the CyQuant GR stock solution in the cell lysate solution at a 1:80 dilution. Subsequently, 500 µL of the working solution were added to each well and placed on a rocker for 5 min at room temperature, protected from light. A total of 200 µL of solution from each well were aliquoted into a clean 96-well black wall clear bottom plate, and fluorescence was read at Ex/Em 480/520 using a Synergy Mx plate reader (Biotek, Winooski, VT, USA).

### 4.10. Statistical Analysis

All experiments were repeated 3 times with N = 3 for each group per experiment. A two-way, repeated measures ANOVA with a Tukey post-hoc test was performed to compare the averages of each experiment between each group at each specific time point for degradation and release studies. Differences in hDF proliferation were measured using a two-way ANOVA.

## 5. Conclusions

Civilian and combat injuries can lead to delays in wound healing and severe complications if not treated correctly. With severity and tissue damage ranging from superficial to chronic, it is important to develop new regenerative therapies capable of delivering drugs and growth factors that promote tissue regeneration over a range of injuries. In this study, we were able successfully to, for the first time, create KSO hydrogels loaded with hPL capable of sustained release of growth factors present in hPL. Our data suggest the hydrogels can be tuned easily by changing KSO and hPL concentrations for the desired mechanical, structural, and growth factor release properties. In addition, the hydrogels were able to promote cell proliferation of hDFs in vitro, demonstrating their suitability as growth factor carrier systems to promote tissue regeneration in future in vivo studies and translation to the clinic. Due to the ease of extracting KSO biomaterials from human hair and their abundance as a renewable, cost-effective resource, KSO biomaterials have great potential in other tissue regenerative therapies.

## Figures and Tables

**Figure 1 ijms-23-04100-f001:**
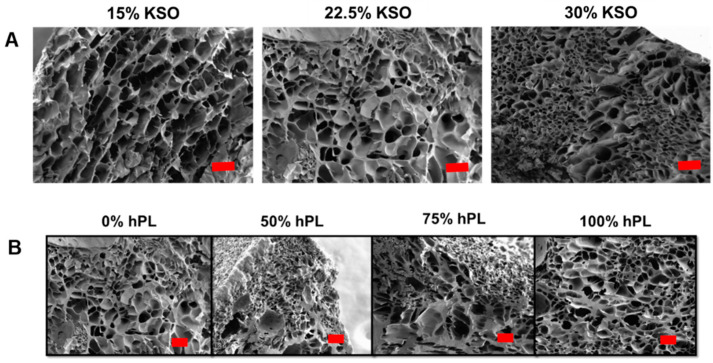
SEM images of human platelet lysate (hPL)-loaded keratin (KSO) hydrogels. KSO hydrogels with no hPL-loading (**A**) showed decrease in porosity with increasing KSO weight percentage. hPL-loaded 22.5% KTN hydrogels (**B**) showed no observable changes in porosity with increase in hPL-loading concentration. Scale bar = 100 µm.

**Figure 2 ijms-23-04100-f002:**
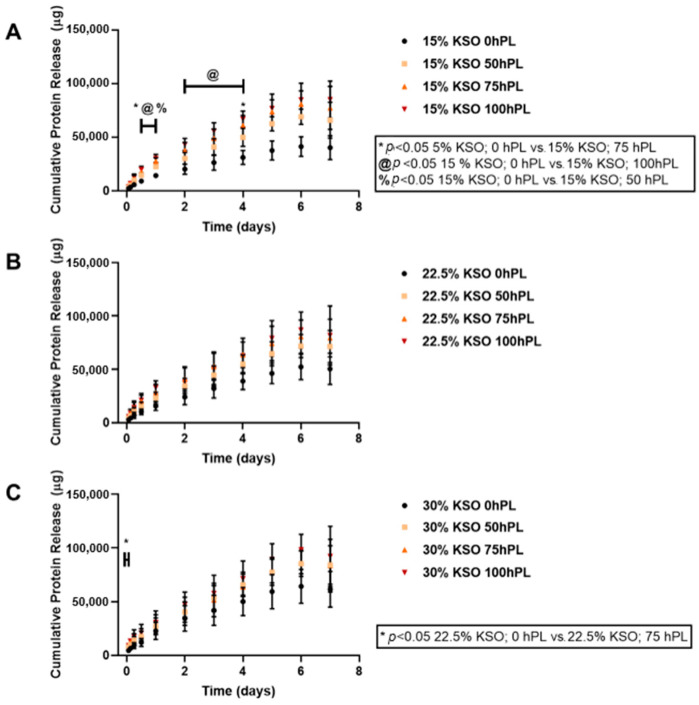
Total protein release against varying hPL-loading concentrations. Total protein release against varying hPL-loading concentrations was quantified by BCA assay and plotted for 15% (**A**), 22.5% (**B**), and 30% (**C**) KSO weight percentage hydrogels. Increase in hPL concentration increased total protein release.

**Figure 3 ijms-23-04100-f003:**
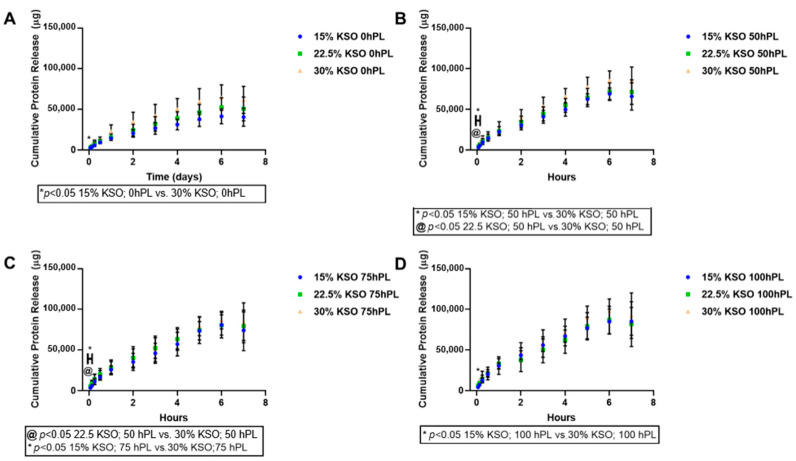
Total protein release against varying KSO hydrogel weight percent. Total protein release against varying KSO hydrogel weigh percent was quantified by BCA assay and plotted for 0% (**A**), 50% (**B**), 75% (**C**), and 100% (**D**) hPL-loading concentration. Increase in KSO hydrogel weight percent decreased total protein release.

**Figure 4 ijms-23-04100-f004:**
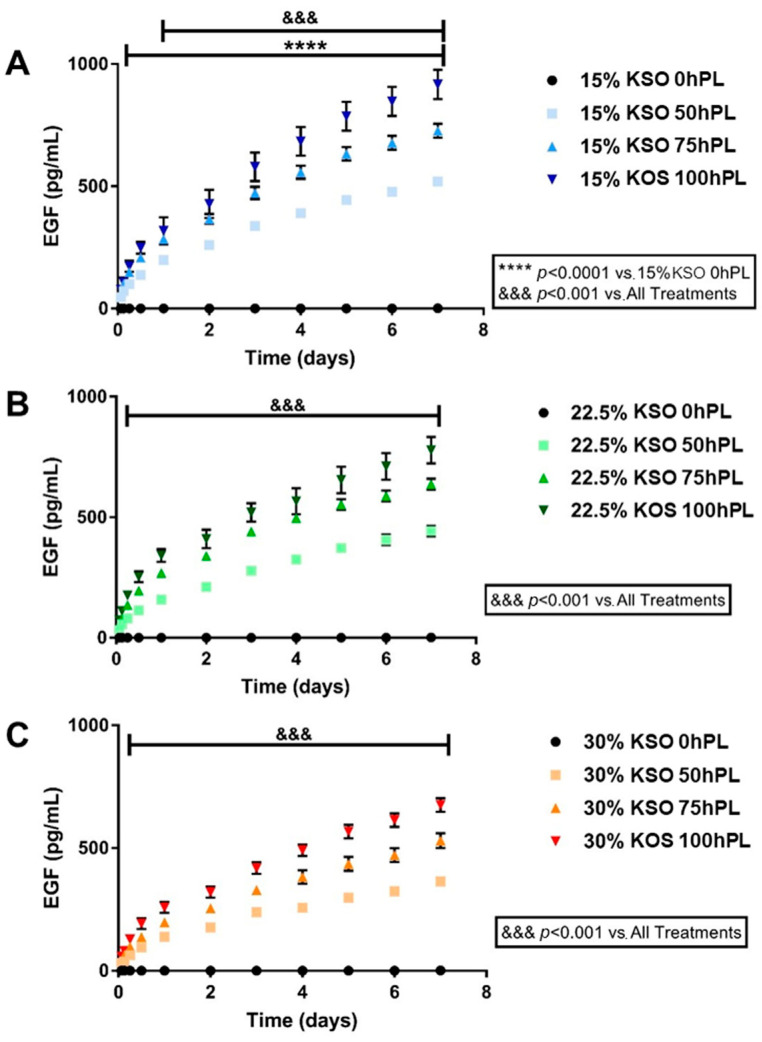
Epidermal growth factor (EGF) release against varying hPL concentrations. EGF release was plotted against varying hPL concentrations for 15% (**A**), 22.5% (**B**), and 30% (**C**) KSO hydrogels over the course of 7 days. For each KSO hydrogel, increase in hPL concentration correlated with increased EGF release. No release was observed with 0 hPL hydrogels for each condition.

**Figure 5 ijms-23-04100-f005:**
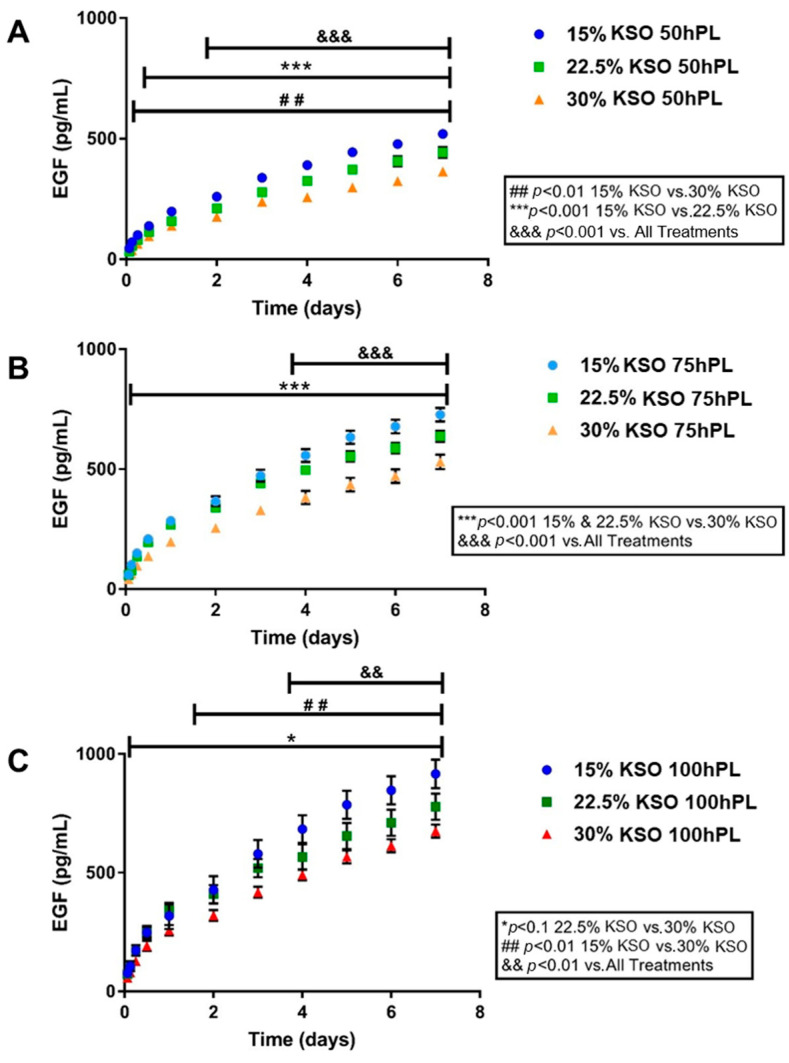
EGF release against varying KSO concentrations. EGF release was plotted against varying KTN concentrations for 50 hPL (**A**), 75 hPL (**B**), and 100 hPL (**C**)—loaded hydrogels over the course of 7 days. For each hPL-loading condition, increase KTN concentration correlated with decrease in EGF release.

**Figure 6 ijms-23-04100-f006:**
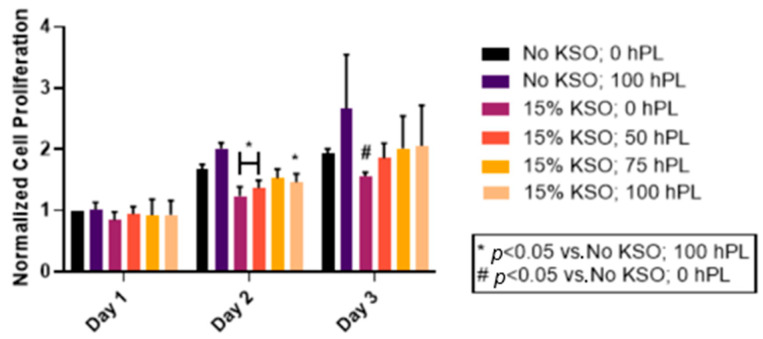
Cell proliferation of hDFs treated with hPL-loaded KSO hydrogels. hDFs were treated with hPL-loaded 15% KSO hydrogels, free-hPL (No KSO; 100 hPL), or left untreated (No KSO; 0 hPL). Cell proliferation was quantified at day 1, 2, and 3 with the CyQuant Cell Proliferation assay.

**Table 1 ijms-23-04100-t001:** Storage (G’), loss (G”), and complex modulus (G*) and loss factor (tan δ) of each hydrogel sample measured by rheological frequency sweep. Results are demonstrated at 1 Hz. Increase in hPL % and keratin (KSO) % (*w*/*v*) decreased and increased the G’, respectively.

Hydrogel Rheological Properties at 1 Hz within the LVER (*n* = 3)					
Keratin (KSO) % (*w*/*v*)	hPL %	G’ (Pa)	G” (Pa)	G* (Pa)	Tan δ
15	0	533.7 ± 40.9	63.48 ± 4.1	537.5 ± 41.0	0.11 ± 0.004
	50	338.5 ± 67.1	44.1 ± 6.2	341.4 ± 67.3	0.13 ± 0.009
	100	240.2 ± 33.3	34.6 ± 5.0	242.7 ± 33.7	0.14 ± 0.000
22.5	0	2079.0 ± 95.6	266.7 ± 32.5	2096.1 ± 99.0	0.13 ± 0.01
	50	1191.1 ± 93.3	150.3 ± 14.6	1200.7 ± 91.9	0.13 ± 0.02
	100	1085.0 ± 108.8	142.0 ± 13.9	1094.3 ± 109.6	0.13 ± 0.000
30	0	6826.0 ± 271.4	999.3 ± 64.0	6898.8 ± 277.4	0.15 ± 0.004
	50	6028.8 ± 709.9	936.1 ± 120.6	6101.1 ± 719.6	0.16 ± 0.004
	100	5926.7 ± 962.0	978.0 ± 191.5	6006.9 ± 980.3	0.16 ± 0.005

## Data Availability

Not applicable.

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
