# Peer review of "Characterization of a Human Platelet Lysate-Loaded Keratin Hydrogel for Wound Healing Applications In Vitro"

_ijms, 2022, doi:10.3390/ijms23084100_

Round 1

Reviewer 1 Report

Journal: International Journal of Molecular Sciences       Manuscript ID: ijms-1653818 

Title: A Human Platelet Lysate-Loaded Keratin Hydrogel for Wound Healing
Applications
In this article, the authors addresses an important topic for the development of new regenerative therapies capable of delivering drugs and growth factors that promote tissue regeneration in a number of injuries. It is based on the keratin protein, which contains cell-binding motifs that promote cell attachment and proliferation.

It is presented for the first time a hydrogel that continuously releases growth factors by promoting cell proliferation of human Dermal Fibroblasts in vitro.

Keratin is a filamentous protein that in a hydrogel supports the release of drugs and promotes wound healing. The present study presents a developed KSO biomaterial that spontaneously forms a hydrogel when rehydrated with Human platelet lysates  that are capable of controlled and sustained release of pro-regenerative molecules.

The authors use various methods of analysis, such as scanning electron microscopy, protein release, quantification of epidermal growth factor, Human platelet lysateс solution and Human platelet lysateс -charged keratin hydrogel treatment, and present in great depth this process.

This article contributes to the expansion of information on the easy extraction of keratin biomaterials from human hair that have potential in other tissue regenerative therapies.

Reviewer 2 Report

In this work, the authors presented a keratin (KSO) biomaterial that spontaneously forms a hydrogel when rehydrated with human platelet lysate (hPL) that are capable of controlled and sustained release of pro-regenerative molecules. The hydrogel showed a tunable release of hPL by changing the KSO hydrogel and hPL-loading concentrations and great biocompatibility for cell proliferation. The authors claimed that the hPL-loaded keratin hydrogels show promise in potential applications for wound healing. However, they had not carried any in vivo experiments. I think this manuscript is not completed, and the authors should carry out the in vivo experiment before submission.

Round 2

Reviewer 2 Report

The authors have not added any animal experiments. In this case, the manucript just provides a new materials for cell culture rather than wound healing. This is far from the theme of the Special Issue.